# Relevance of Targeting the Distal Renal Artery and Branches with Radiofrequency Renal Denervation Approaches—A Secondary Analysis from a Hypertensive CKD Patient Cohort

**DOI:** 10.3390/jcm8050581

**Published:** 2019-04-27

**Authors:** Márcio Galindo Kiuchi, Markus P. Schlaich, Shaojie Chen, Humberto Villacorta, Jan K. Ho, Revathy Carnagarin, Vance B. Matthews, Jocemir Ronaldo Lugon

**Affiliations:** 1Dobney Hypertension Cenre, School of Medicine—Royal Perth Hospital Unit, Faculty of Medicine, Dentistry & Health Sciences, The University of Western Australia Level 3, MRF Building, Rear 50 Murray St, Perth WA 6000, MDBP: M570, Australia; markus.schlaich@uwa.edu.au (M.P.S.); jan.ho@uwa.edu.au (J.K.H.); revathy.carnagarin@uwa.edu.au (R.C.); vance.matthews@uwa.edu.au (V.B.M.); 2Departments of Cardiology and Nephrology, Royal Perth Hospital, Perth WA 6000, Australia; 3Neurovascular Hypertension & Kidney Disease Laboratory, Baker Heart and Diabetes Institute, Melbourne VIC 3004, Australia; 4Cardioangiologisches Centrum Bethanien (CCB) Frankfurt am Main, Medizinische Klinik III, Agaplesion Markus Krankenhaus, Frankfurt am Main 60431, Germany; drsjchen@126.com; 5Cardiology Division, Department of Medicine, Universidade Federal Fluminense, Niterói, RJ 24033-900, Brazil; huvillacorta@globo.com; 6Nephrology Division, Department of Medicine, Universidade Federal Fluminense, Niterói, RJ 24033-900, Brazil; jrlugon@id.uff.br

**Keywords:** hypertension, chronic kidney disease, renal denervation, blood pressure reduction, sympathetic nervous system

## Abstract

We searched for an association between changes in blood pressure (BP) at 12 and 24 months after renal denervation (RDN) and the different patterns of ablation spots placement along the renal artery vasculature. We performed a post-hoc analysis of a 24-month follow-up evaluation of 30 patients who underwent RDN between 2011 and 2012 using our previous database. Patients who had (i) resistant hypertension, as meticulously described previously, and (ii) Chronic kidney disease (CKD) stages 2, 3 and 4. Correlations were assessed using the Pearson or Spearman correlation tests as appropriate. The mean change in systolic ambulatory BP monitoring (ABPM) compared to baseline was −19.4 ± 12.7 mmHg at the 12th (*p* < 0.0001) and −21.3 ± 14.1 mmHg at the 24th month (*p* < 0.0001). There was no correlation between the ABPM Systolic Blood Pressure (SBP)-lowering effect and the total number of ablated spots in renal arteries (17.7 ± 6.0) either at 12 (*r* = −0.3, *p* = 0.1542) or at 24 months (*r* = −0.2, *p* = 0.4009). However, correlations between systolic BP-lowering effect and the number of ablation spots performed in the distal segment and branches were significant at the 12 (*r* = −0.7, *p* < 0.0001) and 24 months (*r* = −0.8, *p* < 0.0001) follow-up. Our findings indicate a substantial correlation between the numbers of ablated sites in the distal segment and branches of renal arteries and the systolic BP-lowering effect in the long-term.

## 1. Introduction

Catheter-based renal denervation (RDN), targeting the renal afferent and efferent sympathetic nerves is an established alternative treatment option for patients with hypertension [1,2,3,4]. The available clinical data suggest that RDN reduces both office and ambulatory blood pressure (BP) in the majority albeit not all treated patients [3,4,5,6,7,8,9]. 

Based predominantly on experimental work in large animal models it has become evident that increasing the number of radiofrequency (RF) lesions in the main renal artery of pigs, although consistently effective in reducing renal norepinephrine (NE) and axon density relative to naive kidneys, this was less reliable in achieving a clear dose-response relationship regarding BP [10]. In contrast, targeted treatment specifically including renal artery branches (segments from renal artery before it reaches the hilum of the kidney), and/or the distal (the last one third) segment of the main renal artery resulted in markedly less variability of response and significantly greater reduction of both NE and axon density than conventional treatment of the main renal artery only. Combination treatment of both the main renal artery and associated branches produced the most significant decline in renal NE and reduction of axon density with least variability of the treatments tested being durable through 28 days post-RDN [10].

The SPYRAL HTN-OFF MED [11] trial, in drug-naïve hypertensive patients, as well as, the SPYRAL HTN-ON MED [12] trial in hypertensive patients on concurrent antihypertensive therapy, both applying treatments in the distal renal arteries and the branches, have demonstrated a convincing and clinically significant reduction of ambulatory BP in comparison to respective sham control groups. Evidence is therefore now available from these consecutive and adequately designed, randomized, sham-controlled trials confirming the BP-lowering efficacy of catheter-based RDN [13].

In the SPYRAL HTN-OFF MED [11] trial, on average 17.9 ± 10.5 ablations were performed in the main renal arteries, and an additional 25.9 ± 12.8 ablations were performed in branch vessels. This is almost 4× the number of ablations (average 11.2 ± 2.8) performed in the SYMPLICITY HTN-3 [6] trial, which purposely spared the branches. Both SPYRAL HTN-OFF MED [11] and SPYRAL HTN-ON MED [12] trials used the same RDN technique and presented similar results in three and six months follow-up, respectively. This data set suggests a broader and focused distal RDN approach is effective in achieving a clinically significant BP-lowering effect in line with pre-clinical results [10].

Previously, our group reported that another different RDN approach was also efficient in reducing BP in resistant hypertensive patients with chronic kidney disease in mid- [14] and long-term [15]. Briefly, a standard irrigated cardiac ablation catheter was inserted into the renal artery and RF applications were performed bilaterally; series of RF pulses aiming >4 ablated spots per renal artery were delivered in a circumferential pullback fashioned from the distal to the proximal renal artery segment. We performed a secondary analysis on a cohort of resistant hypertensive patients with CKD that were previously assessed in our other study which was entitled “Transcatheter Renal Denervation” and published as indicated by [15]. Our current study aims to evaluate whether the RDN technique that we use correlates to BP-lowering or not. Excitingly, our findings indicate that RDN in resistant hypertensive CKD patients provided a significant sustained reduction in BP. More importantly, we observed a clear substantial correlation between the numbers of ablated sites in the distal segment and branches of renal arteries and the long-term lowering of systolic BP. For the first time, we report that the number of treated sites in branches is extremely important for the Systolic blood pressure (SBP)-lowering effect in human CKD patients.

## 2. Methods

We must highlight that this study is a secondary analysis from our previous published data [15]. We conducted a post hoc analysis of a prospective, longitudinal study in 30 hypertensive CKD patients in stages 2, 3, and 4 who underwent RDN. The Ethics Committee of our Hospital approved the study, and all patients signed the informed consent term. Part of this study has previously been published [16].

### 2.1. Study Participants 

We performed a post-hoc analysis of a 24-month follow-up evaluation of 30 patients who underwent RDN from June 2011 to December 2012. Patients were recruited from a university hospital and public health network of the country [15]. The inclusion criteria for this study were as follows: [1] systolic BP ≥160 mmHg (or ≥150 mmHg for patients with type 2 diabetes mellitus), with confirmation using multiple measurements while at the office, despite treatment with nonpharmacologic therapies, i.e., lifestyle modification and the use of at least three antihypertensive medications (including a diuretic) at the maximum doses or confirmed intolerance for medications; [2] estimated glomerular filtration rate (eGFR) determined by the Chronic Kidney Disease Epidemiology Collaboration (CKD-EPI) equation [16] between 15 mL/min/1.73 m^2^ and 89 mL/min/1.73 m^2^ (patients with eGFR >60 mL/min/1.73 m^2^ were required to have microalbuminuria); and [3] age from 18 to 70 years [15].

Exclusion criteria included pregnancy; valvular heart disease with significant hemodynamic consequences; use of warfarin; stenotic valvular heart disease; acute myocardial infarction, unstable angina, stroke, or transient ischemic attack within the previous six months; renovascular anomalies (including renal artery stenosis, angioplasty with or without stenting, or double or multiple main arteries in the same kidney); and diabetes mellitus type 1 or other secondary causes of hypertension. 

### 2.2. Study Procedures and Assessment

Patients underwent a complete medical history and physical examination. Hypertension was diagnosed based on national guidelines in force at that time [17,18]. Patients were screened for other secondary types of hypertension according to guidelines [17,18]. Office BP measurements, 24-h ambulatory BP monitoring (ABPM), screening blood testing, urine samples and echo-Doppler evaluation of the anatomy of the renal arteries was performed or collected as previously described [15]. All patients involved in this study were treated for hypertension for at least 1 year prior to enrolment. Baseline medications were unchanged for at least three months prior RDN to prevent bias in the results, and treatment was maintained at follow-up. We instructed the patients not to change medications or dose after the procedure unless clinically indicated. Drug records for each patient were reviewed and documented at each visit. 

The acute procedures pre- and post-RDN, as well as, anticoagulation, analgesia and anesthesia techniques, were described previously [15]. Angiography of the aorta and renal arteries was performed using an 8-Fr Balkin introducer, and a 7-Fr standard irrigated cardiac ablation catheter was inserted (AlCath Flux eXtra Gold Full Circle 270°; VascoMed GmbH, Binzen, Germany) into the renal artery, allowing RF energy delivery to the renal artery. The catheter was then irrigated, and the length of its gold-tipped electrode (3.5 mm) was approximately four-fold higher than the electrode length (1 mm each one) of the catheter used in SPYRAL HTN-OFF and ON-MED, as shown in Figure 1. RF applications were performed within the renal arteries, bilaterally; a series of RF pulses at 8 W power for 60 seconds each were applied with an irrigation flow rate of 17 mL/min and an aim of >4 RF applications per renal artery, depending on their length. We started circumferential treatment distally and pulled the catheter back towards the proximal renal artery segment (Figure 1). The number of lesions per artery was based on the artery length. 

After the procedure, patients remained hospitalized for 24 h in an inpatient ward. We followed-up patients weekly for the first month, monthly from the second to the sixth months, bimonthly from the seventh to the 12th months, and quarterly during the second year. Doppler ultrasound was performed to evaluate the anatomy of the renal arteries of the patients at the first and sixth months after the RDN. ABPM was performed (with a clinically validated device—CardioMapa; Cardios, São Paulo, Brazil) at the 12th and 24th months after the procedure to evaluate the BP control and the effectiveness of RDN.

### 2.3. Follow-Up

Twenty-seven out of 30 patients completed the 24 months of follow-up, and three patients required chronic renal replacement therapy. One patient-initiated dialysis at the 13th month of follow-up after an acute renal insult associated with pulmonary sepsis. The other two started dialysis at the 14th month, also after acute renal injury episodes: one related to a perforated gastric ulcer and the other following decompensation of heart failure and lung infection. Of note, their eGFR at baseline were 17, 15 and 16 mL/min/1.73 m^2^, respectively. For these three patients, the presented data were collected until their last follow-up visit at 12 months. After these patients began hemodialysis, they were censored from the study. 

### 2.4. Statistical Analysis

Categorical variables (sex, ethnicity, coronary artery disease, atrial fibrillation, stroke, dyslipidemia, smoking, stages of CKD and classes of antihypertensive medications) were expressed as the number of patients (*n*) and percentage (%). Continuous variables (age, body mass index, mean office BP, mean 24-h ABPM, eGFR, number of antihypertensive medications and number of ablated spots) were expressed as mean ± SD, 95% CI or median (Q1–Q3). Comparisons between ABPM measurements at the 12th and 24th month follow-up and their respective baseline values, for systolic and diastolic ABPM, were performed using analysis of variance (ANOVA) for repeated measures and complemented by a post hoc test. Differences between 12- and 24-month follow-up values were compared by the unpaired *t*-test, due to three missing values at the 24th month post-RDN and were expressed as mean and 95% CI. Categorical variable frequencies were compared with the Fisher test. *p* values < 0.05 were considered significant. Correlation between the number of ablated spots per renal arteries regions and ABPM measurements was assessed by the Pearson correlation test, in the case of a Gaussian distribution, or Spearman’s correlation test as an alternative. All statistical analyses were performed using the program GraphPad Prism v 8.0 (GraphPad Software, La Jolla, CA, USA).

## 3. Results

### 3.1. Baseline Characteristics 

Of the 33 initially selected patients, three were excluded because of vascular anomalies that contraindicated RDN. Table 1 displays the general characteristics of the 30 enrolled patients. Seventeen patients were female, and 11 patients had type 2 diabetes mellitus. The mean office systolic/diastolic BP at baseline was 184.9 ± 18.4/106.9 ± 13.3 mmHg, the mean systolic/diastolic ABPM was 152.1 ± 16.6/93.0 ± 11.0 mmHg, the mean eGFR was 61.9 ± 23.9 mL/min/1.73 m^2^, and patients were on an average of 4.6 ± 1.4 different classes of antihypertensive drugs.

### 3.2. Ablation Procedure

In these subjects, we ablated an average of 17.7 ± 6.0 spots in both renal arteries per patient. The number of total lesions delivered, as well as the number delivered in the proximal, middle, and distal segments as well as branches, are summarized in Table 2. There was no difference amongst the mean number of treated sites in proximal, middle and distal sections. However, branches were less frequently ablated in comparison to every other segment. The sum of treated sites in distal portions and branches of renal arteries did not significantly differ from proximal or middle parts (Table 2).

### 3.3. Systolic and Diastolic ABPM-Lowering Effect

The mean change in systolic ABPM compared to baseline was −19.4 ± 12.7 mmHg (95% CI: −24.1 to −14.6) at the 12th month (*p* < 0.0001) and −21.3 ± 14.1 mmHg (95% CI: −26.9 to −15.7) at the 24th month (*p* < 0.0001) follow-up (difference between means −1.9 mmHg, 95% CI: −9.1 to 5.2, *p* = 0.5827), as shown in Figure 2. Similarly, the mean change in diastolic ABPM compared to baseline was −8.4 ± 11.9 mmHg (95% CI: −12.9 to −4.0) at the 12th month (*p* = 0.0015) and −8.1 ± 11.7 mmHg (95% CI: −12.8 to −3.5) at the 24th month (*p* = 0.0013) follow-up (difference between means 0.3 mmHg, 95% CI: −6.0 to 6.5, *p* = 0.9276), as shown in Figure 2.

### 3.4. Correlations between Number of Ablated Spots per Segment and Changes in ABPM

There was no correlation between ABPM systolic blood pressure (SBP)-lowering effect and the total number (17.7 ± 6.0) of ablated spots in renal arteries either at 12 (−19.4 ± 12.7 mmHg; *r* = −0.3, *p* = 0.1631) or at 24 (−21.3 ± 14.1 mmHg; *r* = −0.3, *p* = 0.1316) months post-procedure (Figure 3A,B). When we correlated the 5.2 ± 1.8 treated sites in the proximal segment to the SBP-lowering effect, there was a significant opposite relation at both time points (12th month: *r* = 0.5, *p* = 0.0111, and 24th month: *r* = 0.4, *p* = 0.0332). Correlations between SBP fall and the number of treatments in the middle segment (5.4 ± 2.0) at 12 (*r* = 0.4, *p* = 0.0559) and 24 (*r* = 0.3, *p* = 0.1087) months follow-up were not significant. However, there were moderate correlations between the numbers of ablated sites in distal portion (5.2 ± 2.1) of renal arteries and SBP-lowering effect at 12 (*r* = −0.6, *p* = 0.0004) and 24 (*r* = −0.7, *p* = 0.0002) months follow-up. The same could be observed in branches (1.9 ± 2.9 treated sites) and SBP-lowering effect at 12 (*r* = −0.7, *p* = 0.0001) and 24 (*r* = −0.7, *p* < 0.0001) months follow-up. Surprisingly, the correlations between the number of treated sites in the distal section and branches together (7.1 ± 4.6) became stronger to both follow-up time points (12th month: *r* = −0.7, *p* < 0.0001, and 24th month: *r* = −0.8, *p* < 0.0001, as shown in Figure 3C,D, respectively).

There were no correlations between ABPM diastolic blood pressure (DBP)-lowering effect at 12 (−8.4 ± 11.9 mmHg) and at 24 (−8.1 ± 11.7 mmHg) months post-procedure and the total number of ablated spots (*r* = −0.3, *p* = 0.1542, and *r* = −0.2, *p* = 0.4009, respectively) in renal arteries (Figure 4A,B), proximal segments (*r* = −0.3, *p* = 0.1294 at the 12th month; but there was a positive correlation at the 24th month *r* = 0.4, *p* = 0.0337), and middle segments (*r* = 0.1, *p* = 0.4497, and *r* = 0.3, *p* = 0.1367, respectively). Correlation between DBP fall and the number of treatments in the distal segment (5.4 ± 2.0) was significant at 12 (*r* = −0.5, *p* = 0.0054) and 24 (*r* = −0.4, *p* = 0.0208) months follow-up. For the number of treatments delivered in branches, correlation between DBP fall and the number of treatments was significant at 12 (*r* = −0.5, *p* = 0.0084) and 24 (*r* = −0.4, *p* = 0.0213) months follow-up. A similar correlation was observed for DBP-lowering effect and the number of ablated spots in distal segment summed to branches at 12 (*r* = −0.5, *p* = 0.0031) and 24 (*r* = −0.5, *p* = 0.0029) months follow-up as shown in Figure 4C,D, respectively. 

## 4. Discussion

In this secondary analysis from our previous prospective study, involving a long-term follow-up of resistant hypertensive CKD patient cohort, who underwent RDN with sustained systolic- and diastolic ABPM-lowering effect [15], we found significant correlation between the number of treated sites in the distal segments, branches, and distal segment plus branches of renal arteries and the magnitude of the BP lowering effect.

The average number of ablated spots that we performed in renal arteries per patient resulted in a sizeable systolic ABPM-lowering effect at the 12th and at the 24th month follow-up. When compared to the results presented by the SYMPLICITY HTN-3 at 12 months follow-up in patients who underwent a different range of treated sites in renal arteries, our results were superior only in patients who received up to 13 treatments, at both time points (Table 3). It suggests that a broader range of ablated sites did not interfere in the SBP-lowering effect. Additionally, it corroborates our finding that the total number of treated sites did not correlate to SBP fall magnitude (Figure 3A,B), once the average ablated spots we performed was 18 (Table 3). Comparisons between SYMPLICITY HTN-1 and HTN-2 and our findings were not possible because those trials just used office BP as clinical assessment. Although results from these trials were positive in reducing BP, in SYMPLICITY HTN-1 [19], RDN using a single-electrode catheter reduced NE spillover by 40% on average, but the effect was highly variable, ranging from 0% to 80% [20].

We also compared our findings with results from the SPYRAL HTN-OFF MED [11] and SPYRAL HTN-ON MED [12] trials, as shown in Table 4. The magnitude of the SBP-lowering effect in our study was higher than those reported by the SPYRAL HTN-OFF MED [11] and SPYRAL HTN-ON MED [12] trials, as showed by the differences between means (Table 4). However, caution should be exercised when interpreting such comparison as the time point changes in blood pressure are entirely different. Data presented by us show SBP fall at 12 and 24 months follow-up, while SPYRAL HTN-OFF MED [11] and SPYRAL HTN-ON MED [12] trials reported results from three and six months post-RDN, respectively. In an attempt to compare the number of RF applications delivered by us and the number of ablations performed in those two trials, we analyzed the number of treated sites (Table 5). 

The total number of ablated spots in renal arteries in the SPYRAL HTN-OFF MED [11] and SPYRAL HTN-ON MED [12] trials seemed to be significantly higher compared to ours. However, it did not have any clinical relevance once we did not notice any correlation between the total amount of treated sites and the magnitude of the SBP drop (Figure 3A,B). The same comparison was made between the number of ablations delivered in the main (proximal, middle and distal summed) trunk and branches of renal arteries amongst the three studies. We could observe that the number of lesions delivered by us was significantly smaller when matched to the SPYRAL HTN-OFF MED [11] and SPYRAL HTN-ON MED [12] trials. Pre-clinical data demonstrated that delivery of RF energy to the main renal artery significantly decreased cortical NE concentrations, although a clear dose-response relationship to increasing number of lesions in the main renal artery was not apparent [10]. Regarding branches, our number of ablations was clearly lower in comparison to those performed in the other two trials (Table 5). Pre-clinical data showed that even delivering RF energy to the main renal artery significantly decreased cortical NE concentrations, a clear dose-response relationship to increasing number of lesions [4,8,12] in the main renal artery was not apparent, although heterogeneity in response was reduced [10]. This supports the concept of targeting the distal part of the artery and the branches rather than increasing the number of RF applications in the main renal artery.

Evaluating our data from the perspective that the number of treated sites in branches is extremely important for the SBP-lowering effect, our findings were not surprising as these parameters presented a moderate to strong correlation, even taking in consideration that the number of ablations performed was less numerous than in the SPYRAL HTN-OFF and HTN-ON MED trials. Indeed, it seems to be contradictory, but pre-clinical data reported by Mahfoud and colleagues demonstrated that treating the renal artery branches (only four RF burns) per branch resulted in even greater NE reduction (−82% ± 18%) [10]. Further, the saline solution released by the tip of the standard irrigated cardiac ablation catheter is a variable that can explain the substantial SBP drop size in our study, something not taken into account so far. It is well known that non-irrigated power controlled lesions are smaller and more superficial due to shorter ablation duration and a smaller amount of energy delivered to tissue due to the high peak electrode temperature with a sudden rise in impedance prohibiting ablation [21]. The ablation depth of the currently available RF RDN systems varies between 2 and 4 mm [22,23], thus limiting the accessibility of renal nerves by RF energy delivery in some regions of the renal artery, which suggests that there is an asymmetrical targeting. In contrast, lesions promoted by catheters with irrigated tips are deeper with peak tissue temperatures between 3.5 and 7.5 mm depths [21]. At first glance, it could be irrelevant, as the renal artery wall thickness is thin. Even human renal nerves (48.3%) are mostly placed within 0.5–1.9 mm from the renal arteries lumen, some tick and potentially relevant bundles run through the into the peri-adventitial space and adventitia layer, which are located roughly 1.5–2 mm and >2 mm externally from the vessel lumen, respectively [24]. In addition, pre-clinical data reported that a small piece of gauze soaked in a capsaicin solution wrapped around the renal artery caused selective denervation of afferent nerves (placed mostly externally) and almost abolished the renal content of calcitonin gene-related peptide, eliminating functional response of renal afferents and consequently reducing BP, even in the presence of chemical renal afferents nerves stimulation [25]. Another critical factor is that some renal nerves are surrounded by adipose tissue and lymph nodes [26] protecting them from the RF energy delivered by a conventional ablation method, which can be changed by depth lesions provided by the use of irrigation. Pre-clinical data showed that renal perivascular adipose tissue (PVAT) contains a pool of NE which can be released to alter renal vascular function. PVAT was identified as a mix of white and brown adipocytes, because of the expression of both brown-like (e.g., uncoupling protein 1) and white adipogenic genes. The presence of PVAT (+PVAT) did not alter the response of isolated renal arteries to NE compared to reactions of arteries without PVAT (–PVAT). By contrast, the maximum contraction to the sympathomimetic tyramine was ~2× higher in the artery +PVAT versus –PVAT. The tyramine-induced contraction in + renal perivascular adipose tissue (PVAT) renal arteries was reduced by the α1-adrenoceptor antagonist prazosin and the NE transporter inhibitor nisoxetine. These results suggest that tyramine caused the release of NE from PVAT. RDN significantly (>50%) reduced NE content of renal PVAT but did not modify tyramine-induced contraction of renal +PVAT arteries [26]. Collectively, these data support the existence of a releasable pool of NE in renal PVAT that is independent of renal sympathetic innervation and has the potential to change renal arterial function. These findings also support our theory that deeper lesions provoked by irrigated ablation systems can be a game changer in RDN procedure.

We also found that the sum of ablations performed in distal segment and branches presented a significant correlation to the SBP-lowering effect at the 12th and 24th months after RDN (Figure 3C,D). This effect can be explained by the fact that targeted RF treatment on the distal elements of the renal artery (i.e., the distal segment of the main renal artery and branches placed at the renal artery post-bifurcation) results in significant and relatively uniform reductions in NE and cortical sympathetic axon density and tissue content [10]. Indeed, the greatest magnitude in NE lowering-response (−92% ± 9%) with the least variability was produced with a branch and single-cycle (four RF ablations in the distal segment and four RF burns per branch) treatment in the distal portion of the main trunk of the renal artery [10]. Data about the location and distribution of renal sympathetic nerves is controversial and represents a major underlying substrate for effective RDN procedures. The variation in distribution and density of the renal sympathetic nervous system in 20 human autopsy subjects was recently assessed in detail [27,28]. The largest average number of nerves was observed in the proximal and middle segments of the renal artery and the smallest number in the distal segments [27,28]. On the other hand, another human post-mortem histologic study demonstrated that the number of nerves tended to increase along the length of the artery from proximal to distal segments (proximal = 216; middle = 323; distal = 417) [24]. The mean distance from the lumen to the nerve was longest in the proximal and shortest in the distal segments [27,28]. However, it remains unclear whether anatomical findings can really affect clinical practice in performing the procedure. Even facing this mismatch, we think that an appropriate strategy would be to target the distal renal artery in order to achieve successful denervation of renal afferent and efferent nerves, as our findings revealed (Figure 3C,D).

Another crucial point is that our patients presented CKD beyond hypertension, which was not observed in the SYMPLICITY HTN-1, HTN-2 and HTN-3 trials neither in the SPYRAL HTN-OFF and HTN-ON MED trials. As we know, the sympathetic activation is a hallmark of the essential hypertensive state occurring early in the clinical course of the disease [29,30]. In CKD, the sympathetic overactivity appears to be manifested at the earliest clinical stage of the disease, being directly related to the severity of the renal failure state [31,32,33]. In both conditions, hypertension and renal failure, the mechanisms of the hyperadrenergic state are manifold and include reflex and neurohumoral pathways [29,30,34]. The blunting of this sympathetic hyperactivity from both conditions and the potential interruption of the feedback loop of the renin-angiotensin-aldosterone system may, at least in part, account for our increased and sustained BP-lowering effect.

## 5. Study Limitations

The relatively small sample is a limitation of this study. However, the present post hoc analysis is unique in the literature as it addresses the correlation between the number and location of RDN ablations and the sustained BP-lowering effect in the long-term. Nevertheless, because this analysis was performed from an uncontrolled study, our findings should be interpreted with caution. In addition, this is a post hoc analysis and, as such, the primary research [15] was not designed for this aim. 

## 6. Conclusions

Our findings indicate that RDN in resistant hypertensive CKD patients provided a significant sustained reduction in BP. More importantly, we observed a clear and substantial correlation between the numbers of ablated sites in the distal segment and branches of renal arteries and the systolic BP-lowering effect in the long-term. Although encouraging, our data are from a post hoc analysis and must be validated in a larger and properly designed study to address this specific aim.

## 7. Impact on Daily Practce

The positive results from SPYRAL HTN-OFF MED, RADIANCE SOLO and SPYRAL HTN-ON MED signify re-ignited the RDN field. In concordance of these studies, our data corroborate the perspective that the number of treated sites in branches is extremely important for the SBP-lowering effect.

## Figures and Tables

**Figure 1 jcm-08-00581-f001:**
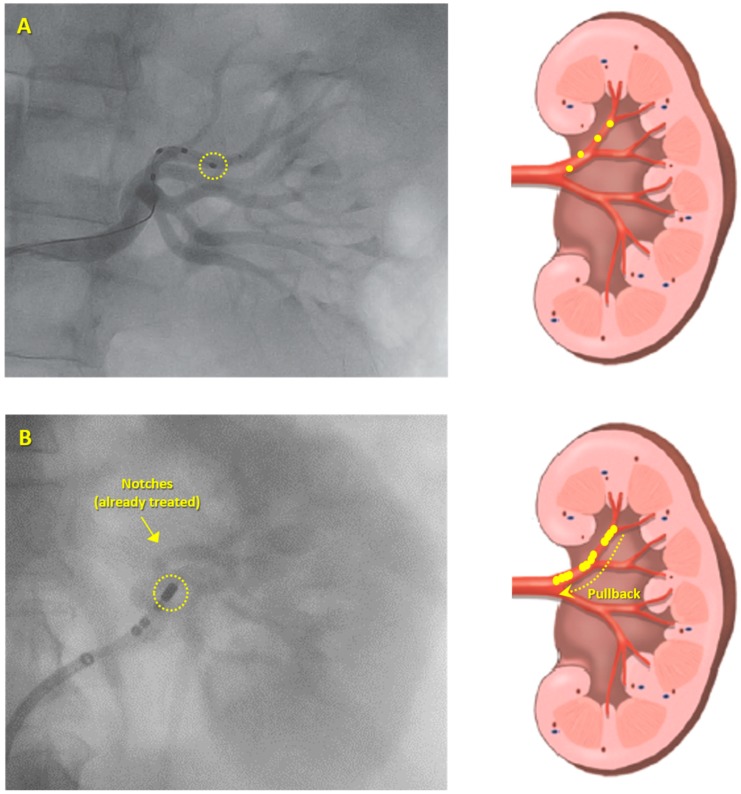
Types of renal denervation approach used. (**A**) Symplicity Spyral multielectrode renal denervation (RDN) catheter (Medtronic, Galway, Ireland) which allows simultaneous or sequential energy treatments into a left renal artery (each one out of four electrodes is 1 mm in length). (**B**) 7-Fr standard irrigated cardiac ablation catheter with a 3.5 mm electrode tip length (AlCath Flux eXtra Gold Full Circle 270°; VascoMed GmbH, Binzen, Germany) was inserted into a left renal artery and was moved in a circumferential pullback fashioned from the distal to the proximal renal artery segment. Yellow dots represent ablated spots.

**Figure 2 jcm-08-00581-f002:**
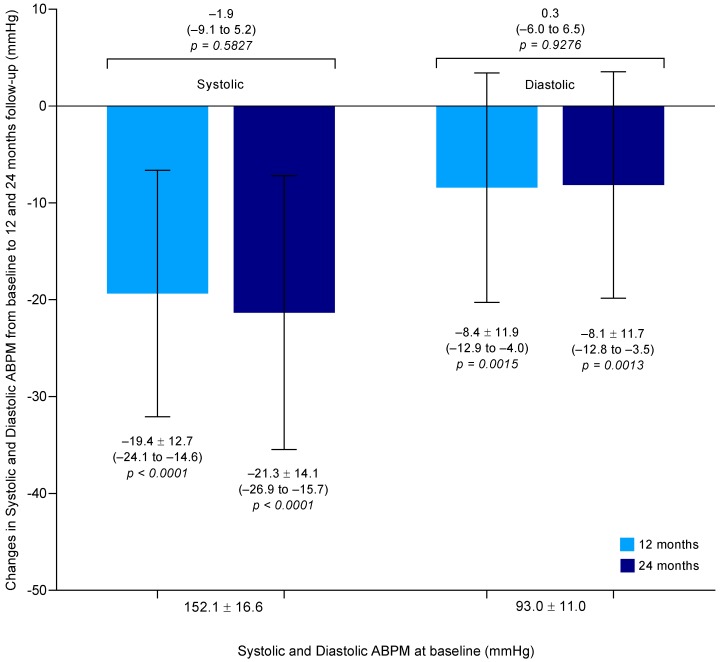
Changes at 12 and 24 months of follow-up in ambulatory systolic blood pressure and diastolic blood pressure post-RDN. Data are mean ± SD (95% CI). ABPM = 24-h ambulatory blood pressure monitoring.

**Figure 3 jcm-08-00581-f003:**
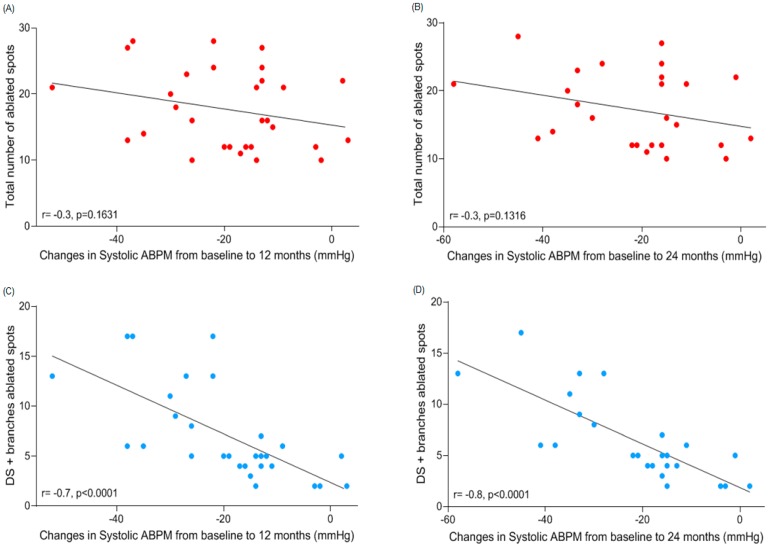
Correlations between change from baseline to 12 and 24 months follow-up in ambulatory systolic blood pressure after renal denervation and (**A**,**B**) number of ablated spots in total (red dots) and (**C**,**D**) distal segment summed to branches (blue dots) of renal arteries. ABPM = ambulatory blood pressure monitoring. DS = distal segment. Correlation for the number of treatments performed in the distal segment and branches were significant at both time points.

**Figure 4 jcm-08-00581-f004:**
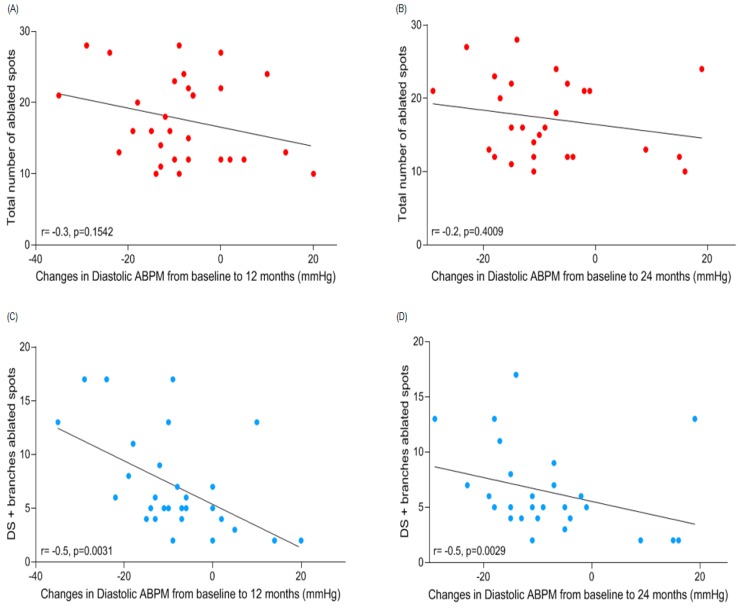
Correlations between changes from baseline to 12 and 24 months of follow-up in ambulatory diastolic blood pressure after renal denervation and (**A**,**B**) number of ablated spots in total (red dots) and (**C**,**D**) distal segment summed to branches (light green dots) of renal arteries. ABPM = ambulatory blood pressure monitoring. DS = distal segment. Correlations for the number of treatments performed in the distal segment and branches were significant at both time points.

**Table 1 jcm-08-00581-t001:** General features of patients at baseline.

Variables	Values
*n*	30
Age (years)	55 ± 10
Female sex (%)	17 (57%)
Ethnicity (non-white) (%)	21 (70%)
Body mass index, kg/m^2^	30.8 ± 4.9
Coronary artery disease (%)	5 (17%)
Atrial fibrillation (%)	2 (7%)
Stroke (%)	6 (20%)
Type 2 diabetes (%)	11 (37%)
LDL-cholesterol >130 mg/dL (%)	19 (63%)
Smoking (%)	3 (10%)
Mean office systolic/diastolic blood pressure, mmHg	184.9 ± 18.4/106.9 ± 13.3
Mean 24-h systolic/diastolic ABPM, mmHg	152.1 ± 16.6/93.0 ± 11.0
eGFR, mL/min/1.73 m^2^ (CKD-EPI)	61.9 ± 23.9
Stages of CKD 2/3/4	19 (63%)/6 (20%)/5 (17%)
Number of antihypertensive medications (%)	4.6 ± 1.4
ACE-I	5 (17%)
ARB	25 (83%)
Aliskiren	3 (10%)
α-1 blocker	1 (3%)
β-blocker	24 (80%)
Clonidine or Moxonidine	11 (37%)
Calcium channel blocker	25 (83%)
Diuretic	30 (100%)
Aldosterone antagonist	3 (10%)
Vasodilator	4 (13%)

Data are mean ± SD or *n* (%). ACE-I = angiotensin converting enzyme inhibitors. ARB = angiotensin-receptor blockers. BMI = body-mass index. eGFR = estimated glomerular filtration rate. LDL = Low Density Lipoproteins. ABPM = Ambulatory Blood Pressure Monitoring. CDK-EPI = the Chronic Kidney Disease Epidemiology Collaboration equation.

**Table 2 jcm-08-00581-t002:** Number of ablated spots in renal arteries (left + right) in 30 patients.

	Mean ± SD	Difference between Means (95% CI)			
**Total**	17.7 ± 6.0	-	-	-	-
		PS vs. MS	PS vs. DS	PS vs. Br	PS vs. DS + Br
**Proximal segment (PS)**	5.2 ± 1.8	0.2 (−04 to 0.8) *p* = 0.9039	0.0 (−1.5 to 1.5) *p* > 0.9999	3.3 (1.4 to 5.3) *p* = 0.0002	1.9 (−0.9 to 4.6) *p* = 0.3227
			MS vs. DS	MS vs. Br	MS vs. DS + Br
**Middle segment (MS)**	5.4 ± 2.0	-	0.2 (−1.3 to 1.7) *p* = 0.9984	3.5 (1.5 to 5.5) *p* = 0.0001	1.7 (−1.1 to 4.4) *p* = 0.4626
				DS vs. Br	DS vs. DS + Br
**Distal segment (DS)**	5.2 ± 2.1	-	-	3.3 (2.2 to 4.5) *p* < 0.0001	1.9 (0.3 to 3.5) *p* = 0.0144
					Br vs. DS + Br
**Branches (Br)**	1.9 ± 2.9	-	-	-	5.2 (4.0 to 6.4) *p* < 0.0001
**Distal segment (DS) + branches (Br)**	7.1 ± 4.6	-	-	-	-

**Table 3 jcm-08-00581-t003:** Number of ablated spots in renal arteries and changes in systolic ABPM after renal denervation.

	Our Findings ^a^		SYMPLICITY HTN-3 ^b^				
	12 Months	24 Months	12 Months				
***n***	30	27	31	31	32	32	24
**Number of ablations**	average 18		≤9	10	11	12 or 13	≥14
**Changes in systolic ABPM from baseline, mmHg**	−19.4 ± 12.7 *p* < 0.0001	−21.3 ± 14.1 *p* < 0.0001	−2.8 ± 10.8 *p* = 0.02	−6.9 ± 15.5 *p* = 0.78	−0.4 ± 18.2 *p* = 0.77	−9.3 ± 9.6 *p* = 0.002	−12.2 ± 19.1 *p* = 0.16
**Difference between means, mmHg (95% CI)**			−16.6 (−26.1 to −7.1) *p* < 0.0001	−12.5 (−22.0 to −3.0) *p* = 0.0046	−19.0 (−28.4 to −9.6) *p* < 0.0001	−10.1 (−19.5 to −0.7) *p* = 0.0299	−7.2 (−17.3 to 2.9) *p* = 0.2526
**Our findings at 12 months vs. SYMPLICITY HTN-3**							
**Difference between means, mmHg (95% CI)**			−18.5 (−28.3 to −8.7) *p* < 0.0001	−14.4 (−24.3 to −4.6) *p* = 0.0014	−20.9 (−30.7 to −11.1) *p* < 0.0001	−12.0 (−21.8 to −2.2) *p* = 0.0098	−9.1 (−19.6 to 1.4) *p* = 0.1116
**Our findings at 24 months vs. SYMPLICITY HTN-3**							

Data are mean ± SD and 95% CI. ^a^ Catheter used: standard irrigated cardiac catheter ablation—AlCath Flux eXtra Gold Full Circle 270° (3.5 mm tip length: VascoMed GmbH, Binzen, Germany). ^b^ Catheter used: Symplicity™ Renal Denervation System (1 mm tip length: Medtronic, Inc., Santa Rosa, CA, USA).

**Table 4 jcm-08-00581-t004:** Changes in systolic ABPM after renal denervation.

			SPYRAL HTN	
	Our Findings ^a^		OFF MED ^b^	ON MED ^b^
	12 Months	24 Months	3 Months	6 Months
***n***	30	27	37	36
**Changes in systolic ABPM from baseline, mmHg**	−19.4 ± 12.7 (−24.1 to −14.6) *p* < 0.0001	−21.3 ± 14.1 (−26.9 to −15.7) *p* < 0.0001	−5.5 ± 13.9 * (−9.1 to −2.0) *p* = 0.0031	−9.0 ± 11.0 (−12.7 to −5.3) *p* < 0.0001
**Difference between means, mmHg (95% CI)**			−13.9 (−22.2 to −5.6) *p* = 0.0001	−10.4 (−18.7 to −2.1) *p* = 0.0078
**Our findings at 12 months vs. SPYRAL HTN**				
**Difference between means, mmHg (95% CI)**			−15.8 (−24.3 to −7.3) *p* < 0.0001	−12.3 (−20.9 to −3.7) *p* = 0.0016
**Our findings at 24 months vs. SPYRAL HTN**				

Data are mean ± SD and 95% CI. ^a^ Catheter used: standard irrigated cardiac catheter ablation—AlCath Flux eXtra Gold Full Circle 270° (3.5 mm tip length: VascoMed GmbH, Binzen, Germany). ^b^ Catheter used: Symplicity Spyral multielectrode RDN catheter four electrodes, 1 mm length each one, which allows simultaneous or sequential energy treatments: Medtronic, Galway, Ireland). * SD calculated from the formula SE = SD/$\sqrt{n}$, in this case SE = 2.3, SD = unknown, *n* = 37 (2.3 = SD/$\sqrt{37}$ → SD = 13.9).

**Table 5 jcm-08-00581-t005:** Number of ablated spots in renal arteries.

		SPYRAL HTN	
	Our Findings ^a^	OFF MED ^b^	ON MED ^b^
**Total**	17.7 ± 6.0	43.8 ± 11.1	45.9 ± 13.7
**Difference between means (95% CI)** **Our findings vs. SPYRAL HTN**	-	−26.1 (−32.5 to −19.7) *p* < 0.0001	−28.2 (−34.7 to −21.7) *p* < 0.0001
**Ablations in the main artery ** **(proximal, middle and distal segments)**	5.3 ± 2.0	17.9 ± 10.5	19.3 ± 8.9
**Difference between means (95% CI)** **Our findings vs. SPYRAL HTN**	-	−12.7 (−17.5 to −7.9) *p* < 0.0001	−14.1 (−19.0 to −9.2) *p* < 0.0001
**Ablations in branches**	1.9 ± 2.9	25.9 ± 12.8	26.6 ± 11.7
**Difference between means (95% CI)** **Our findings vs. SPYRAL HTN**	-	−24.0 (−30.1 to −17.9) *p* < 0.0001	−24.7 (−30.9 to −18.6) *p* < 0.0001

Data are mean ± SD and 95% CI. ^a^ Catheter used: standard irrigated cardiac catheter ablation—AlCath Flux eXtra Gold Full Circle 270° (3.5 mm tip length: VascoMed GmbH, Binzen, Germany). ^b^ Catheter used: Symplicity Spyral multielectrode RDN catheter (four electrodes, 1 mm length each one, which allows simultaneous or sequential energy treatments: Medtronic, Galway, Ireland).

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
