# Peer review of "Relevance of Targeting the Distal Renal Artery and Branches with Radiofrequency Renal Denervation Approaches—A Secondary Analysis from a Hypertensive CKD Patient Cohort"

_jcm, 2019, doi:10.3390/jcm8050581_

Reviewer 1 Report

Overall this is a very well done analysis of existing data regarding RDN in subjects with CKD 2-4. There are some minor clarifications that should be addressed. The findings suggest that a prospective trial in CKD patients with resistant hypertension should be considered.

Critique

1.      The Abstract  lines 32-33 should specify the stages of CKD of the subjects in the study (i.e., CKD 2, 3 and 4).

2.      The Introduction is well written and provides appropriate background for the current study.

3.      Methods, line 104, should specify the lower limit of microalbuminuria (e.g., ≥ 30 mg/g creatinine or ≥ 300 mg/g creatinine). The first is stage A2 and the second A3. In other words, how severe did the albuminuria need to be if the eGFR was > 60 ml/min/1.73m2 for subjects to be included?

4.      Methods, lines 111-112 and 119-120: The statements are redundant. Please delete one of the repeated sentence/phrases.

5.      Figure 1: It may be advantageous to use a color other than green for the ablated spots especially as they are on a red background. It may difficult for color blind readers to appreciate the spots.

6.      I understand that the follow up lines 155-162 are included in Methods since these were censored post hemodialysis initiation, but could equally be put into the Results. Also, re: these subjects the Discussion and Table 1: A comment by the authors as to their view, if any, re: the patients who ended up on dialysis beginning in CKD 4 as a baseline even prior to their acute kidney injury would be appreciated. This is notable since these 3 individuals are from the cohort of only 5 patients with CKD 4.

7.      Discussion lines 261 and others:  The actual numbers are included in the Results and Tables and need not be repeated here. The Discussion can thus be streamlined.

8.      Discussion lines 320-321: Firstly, the dimension should be “mm” not “cm” from the vessel lumen. Secondly, the study by Atherton (Ref. #24) reported that 15.5% of the renal nerves were 1.5-2.0 mm and 9.5% were 2.0-2.5 mm from the lumen, totaling 25% which is not therefore where the nerves are “mostly placed.” Although the concept being discussed is supported, the comment as written is not precisely accurate as “most” or 48.3% of the nerves were in the 0.5-1.9 mm ring from the lumen.  This should be corrected.

9.      Realizing these were post hoc analyses and the subjects were on medications, yet were plasma renin levels or urine norepinephrine levels obtained in any of these subjects? These would have been valuable data to have.

10.   What was the change, if any, in eGFR from baseline to the end of the study? This would be very helpful to know and should be available since CKD patients were very likely to have been followed for their renal function over this 2 year period.

Other

Line 32: “meticulous” should be “meticulously”

Line 275:  “confronted” the findings; is “compared” meant?

Line 296:  “expressively” should be “expressly”

Line 306:  “amount of ablations” should be “number of ablations”

Author Response

Please find attached the file.

Reviewer 2 Report

The manuscript reported the association of the number of ablated sites in the distal segment and branches of renal arteries with the long-term BP-lowering effect. However, the referential significance of the study is limited as the sample size is too small compared to similar kinds of studies. The findings in the manuscript seem not novel considering the existing knowledge and literature.  Moreover, 24-months is not considered as the ‘long-term’ in the RDN procedure, especially considering the data was collected years back and the follow-up track beyond 24-months should be available.

Author Response

Please find attached the file.

Reviewer 3 Report

In present study by Kiuchi MG et al examined the association of blood pressure and renal denervation. They studied 30 patients and followed 24 month after RDN. RND were perfumed in renal artery, distal segment, and branches. Author found no correlation of SBP and number of ablated spot in renal arteries but found significant correlation in distal segment and branches using post-hoc analysis.  Manuscript is meaningful and well needed for HTN audience. I have few comments.

1.     Promising results observed in initial SIMPLICITY trials, renal denervation failed to achieve its efficacy end points as a treatment for resistant hypertension in the SYMPLICITY HTN-3 trial, the largest trial of this treatment to up to date. Unfortunately, EnligHTN IV Trial study prematurely stopped. Many RDN were terminated in the United states than completed.

2.     What are the 13 treatment? I could not understand why this method is superior. Could author discuss why artery has not effect but branches and distal segments in cellular level?

3.     The differences in reducing or not reducing blood pressure is device dependent? Between the trails.

4.     Author should consider defining difference between branches and distal segments.

5.     Did all the RDN were successful? Any cases of pseudoaneurysms?

6.     Why does author call it as prospective study?

Author Response

Please find attached the file.

Round  2

Reviewer 2 Report

Thank you for the response to my comments. The major findings have been published in the previous two articles by the same first author, including the same patients, detailed procedures and the 24-month follow-up. I understand the current manuscript focused on the correlation of the number of ablated sites with long-term BP, but if the author thought this was a significant finding, it should have been pointed out in the previous reports. In my opinion, the current findings are not novel and significant enough.

I don’t agree with the author’s claim that “the present series is the largest in the literature to address percutaneous renal artery denervation in CKD patients and it has the longest follow-up.”, as in similar clinical studies using renal denervation (e.g., Eur Heart J. 2014 Jul;35(26):1752-9.), the follow-up was available at 36 months including 40 subjects. Considering that the results of the 24-month follow-up was already published (J Clin Hypertens (Greenwich). 2016 Mar;18(3):190-6.), I expected the author to have >24 month to increase the significance and novelty of the current study. 

Author Response

1. We appreciate the reviewer's comments and acknowledge her/his reservations. We would, however, like to make the following points relevant to the current context: While we have previously reported on this cohort, it was not until more recently (ref) that the relevance of distal vs proximal ablation has become apparent as demonstrated in both animal work (swine ref JACC) and humans (J Hypertens. 2017 Feb;35(2):369-375.) These findings prompted us to investigate whether differences in the ablation pattern, i.e. distal vs proximal may also have influenced our results. Indeed, findings from or post-hoc analysis are in line with a more pronounced BP lowering effect with distal ablation. While confirmatory, we believe this is clinically highly relevant information in a vulnerable patient cohort and adds further support for a distal approach future studies and clinical management of patients. The retrospective nature of our analysis in this specific context where ablation spots were chosen more randomly is not necessarily a disadvantage to support the concept. 

2. The paper by Esler et al. (Eur Heart J 2014) was a 3 year follow up of the Symplicity-HTN 2 trial. This was not a CKD study per se but did include patients with eGFR in respective ranges. We acknowledge the reviewer's comment and have no rephrased this section:

 "the present series is one of the largest in the literature ....with a complete follow up to 2 years."

 Of note, Ethics approval for this study was only obtained for a maximum of 2 years follow up. Hence longer-term FU was not feasible.